# Copy number rather than epigenetic alterations are the major dictator of imprinted methylation in tumors

Alex Martin-Trujillo[1], Enrique Vidal [2,3], Ana Monteagudo-Sánchez[1], Marta Sanchez-Delgado [1], Sebastian Moran [4], Jose Ramon Hernandez Mora[1], Holger Heyn[3,5], Miriam Guitart[6], Manel Esteller[4,7,8] & David Monk[1]

It has been postulated that imprinting aberrations are common in tumors. To understand the role of imprinting in cancer, we have characterized copy-number and methylation in over 280 cancer cell lines and confirm our observations in primary tumors. Imprinted differentially methylated regions (DMRs) regulate parent-of-origin monoallelic expression of neighboring transcripts in *cis*. Unlike single-copy CpG islands that may be prone to hypermethylation, imprinted DMRs can either loose or gain methylation during tumorigenesis. Here, we show that methylation profiles at imprinted DMRs often not represent genuine epigenetic changes but simply the accumulation of underlying copy-number aberrations (CNAs), which is independent of the genome methylation state inferred from cancer susceptible loci. Our results reveal that CNAs also influence allelic expression as loci with copy-number neutral loss-of-heterozygosity or amplifications may be expressed from the appropriate parental chromosomes, which is indicative of maintained imprinting, although not observed as a single expression foci by RNA FISH.

[1] Imprinting and Cancer group, Cancer Epigenetic and Biology Program (PEBC), Institut d'Investigació Biomedica de Bellvitge (IDIBELL), Avinguda Granvia, L'Hospitalet de Llobregat, 08907 Barcelona, Spain. [2] Centre for Genomic Regulation (CRG), The Barcelona Institute of Science and Technology, Barcelona, Spain. [3] Universitat Pompeu Fabra (UPF), Barcelona, Spain Universitat Pompeu Fabra (UPF), 08003 Barcelona, Spain. [4] Cancer Epigenetics group, Cancer Epigenetic and Biology Program (PEBC), Institut d'Investigació Biomedica de Bellvitge (IDIBELL), Avinguda Granvia, L'Hospitalet de Llobregat, 08907 Barcelona, Spain. [5] CNAG-CRG, Centre for Genomic Regulation (CRG), Barcelona Institute of Science and Technology (BIST), Baldiri i Reixac 4, 08028 Barcelona, Spain. [6] Genetics Laboratory, UDIAT- Diagnostic Centre, Corporació Sanitària Parc Taulí, 08208 Sabadell, Spain. [7] Department of Physiological Sciences II, School of Medicine, University of Barcelona, Barcelona 08907 Catalonia, Spain. [8] Institucio Catalana de Recerca i Estudis Avançats (ICREA), 08010 Barcelona, Spain. Alex Martin-Trujillo and Enrique Vidal contributed equally to this work. Correspondence and requests for materials should be addressed to D.M. (email: dmonk@idibell.cat)

Cancer is the leading cause of death in adults. Tumors themselves are typically classified using pathological criteria and tissue origin. However following the huge investment from large international consortia, cancers can now be sorted into molecular subgroups based on genetic and epigenetic profiles[1] which has led to targeted therapies for subtypes with specific alterations.

To help with molecular classification, we have investigated whether aberrations at imprinted loci can stratify tumors from different tissue origins.

Genomic imprinting is the parent-of-origin specific mono-allelic transcription, regulated in part by allelic difference in DNA methylation established in the male and female germline and maintained throughout somatic development[2]. In addition to being indispensible for growth, imprinted genes have been suggested to play a crucial role in driving oncogenic switch or suppressing tumor development[3]. Deregulated expression, which includes the reactivation of the normally silent allele (commonly referred to as loss-of-imprinting, LOI) or the silencing of the transcribed allele, has been implicated in childhood cancer associated with the classical imprinting disorder Beckwith–Wiedemann syndrome[4]. In these cases gains of methylation at the paternally methylated H19 DMR or paternal uniparental disomy (copy-number neutral loss-of-heterozygosity, cnnLOH) result in over-expression of IGF2-miR483 and the concomitant silencing of H19-miR675[5]. However it is becoming evident that an increasingly number of imprinted genes shows irregularities in adult tumors. For example aberrant expression of transcripts in the H19-IGF2 domain has been reported in more than 15 tumor types, with somatic amplifications of IGF2 and miR483 implicated in the initiation of colorectal cancer[5, 6], with copy-number gains being mutually exclusive with P13K pathway-activating mutations[7]. In addition over-expression of H19-miR675 is frequently observed in colorectal cancers, which suppress the expression of the tumor-suppressor RB1[8]. Downregulation of DIRAS3 (also known as ARH1) by either deletion or methylation is frequently observed in breast and ovarian cancers[9, 10] and has been implicated in sensitivity to Cisplatin, a widely used platinum-based chemotherapy agent[11]. Furthermore epigenetic silencing and deletions of L3MBTL1, a member of the polycomb-like family located at chromosome 20q, is associated with hematological malignancies showing increased genome instability[12, 13].

Imprinted deregulation has also helped modify our thoughts on the classical genetic events leading to cancer. For example, despite the RB1 gene being at the center of Knudson's two hit hypothesis in which familial retinoblastoma displays autosomal dominant inheritance with the first hit present in the germline and the second somatically acquired[14], this gene is preferentially expressed from the maternal chromosome[15]. This is evident in epidemiological data from sporadic retinoblastoma patients in which the maternal chromosome 13 is preferentially lost[16].

Ignoring placenta-specific imprinted genes[17–19], there are currently ~150 transcripts in the human genome mapping to ~30 domains suggesting that other imprinted genes may contribute to the development of cancer in adults. However to date very few studies have systematically looked at the influence of somatically acquired copy-number alterations (CNAs) on imprinted methylation profiles in cancer. We have performed an analysis to determine the association between CNAs and DNA methylation profiles for three cancer cell types originating in the lung, colon and breast which rank among the most frequent solid tumor types, as well as liver, which is a prevalent cause of cancer-associated deaths due to hepatitis B virus infections (https://www.cancer.org/research/cancer-facts-statistics). Overall we observe that all four-tumor types show highly aberrant profiles, with a high proportion of aberrant imprinted DMR methylation patterns being associated with cancer-associated CNAs and not somatically acquired epigenetic defects.

## Results

We utilize copy-number and epigenetic data from a large set of cancer cell lines from the Sanger Institutes COSMIC project[20, 21] with observations confirmed in cohorts of well-characterized primary cancer samples from The Cancer Genome Atlas (TCGA) consortium[7, 22–25]. To determine the interplay between CNAs and DNA methylation we combined data from genotyping (Affymetrix Genome-wide SNP6 arrays) and methylation (Illumina Infinium HumanMethylation450 BeadChips, HM450k) platforms that interrogated 906,000 single-nucleotide polymorphisms (SNPs) and 485,577 CpG dinucleotides, respectively. Our group has previously characterized the Infinium HM450k methylation array for parent-of-origin methylation associated with imprinted regions[17]. In this study we focused on the 661 probes mapping to 37 imprinted DMRs.

**CNAs at imprinted domains in cancer cell lines.** High-resolution copy-number data for each imprinted domain was obtained from COSMIC cell line project that included 173 lung, 50 colorectal, 49 breast and 15 hepatocarcinoma cell lines. We confirmed the CNAs using other low-resolution molecular cytogenetic methods, including SKY karyotyping and standard DNA fluorescence in situ hybridization (FISH) (Fig. 1a–c)(http://www.pawefish.path.cam.ac.uk)[25]. We identify thousands of aberrations including deletions, amplifications and cnnLOH. Copy-number calling was reported as total copy number and minor allele, where a normal diploid state is 2:1 and cnnLOH having a total copy number $\geq 2$ and a minor allele count of zero, an event in which one allele is deleted and the remaining one amplified. In all cases an estimated ploidy baseline (total alleles/baseline ploidy) was also calculated for each cell line (Supplementary Data 1 and Supplementary Figs 1–6) since total copy number $\geq 2$ could represent amplification in a diploid tumor but a deletion in a hyperploid tumor. No cell lines had normal copy-number at all imprinted domains, with the least affected being the breast-derived CAL-51 cell line that was normal except for cnnLOH of the chromosome 11q15 domains (2:0) and the colorectal cell line LS-180 which had isolated cnnLOH of GRB10 (2:0). The most severe CNAs were observed in lung cancer cell lines, with the PPIEL locus on chromosome 1 being amplified up to 14 times in DMS-273 (14:1) NCI-H378 (14:1) and NCI-H1836 (14:2) with the NCI-H1694 cell line having all copies of one allele (14:0). Such huge gains were not restricted to the PPIEL locus as the MCTS2/HM13 domain was also amplified 14 times in the SKLU-1 cell line (14:2). The full catalog of copy-number and methylation results is available at www.humanimprints.net (Supplementary Data 2).

Next, we wish to determine the size of the CNAs since telomere/centromere bound aberrations tend to be larger than focal internal CNAs[24]. Indeed we observe that focal internal CNAs incorporating imprinted domains were smaller in 96% of cases and were in higher copy number 79% of the time. For example the average telomere bound amplification for HTR5A on chromosome 7 is 41 Mb, with an average total copy-number of 4.4, whereas internal amplifications are on average 8.5 Mb with >9 copies (Supplementary Data 3). For chromosomes harboring more than one imprinted domain, the CNAs may be focal or involve the entire chromosome arm (Fig. 1c, d)(extended data available at the www.humanimprints.net). This was confirmed using DNA FISH in the breast cancer cell line HCC1954 that has

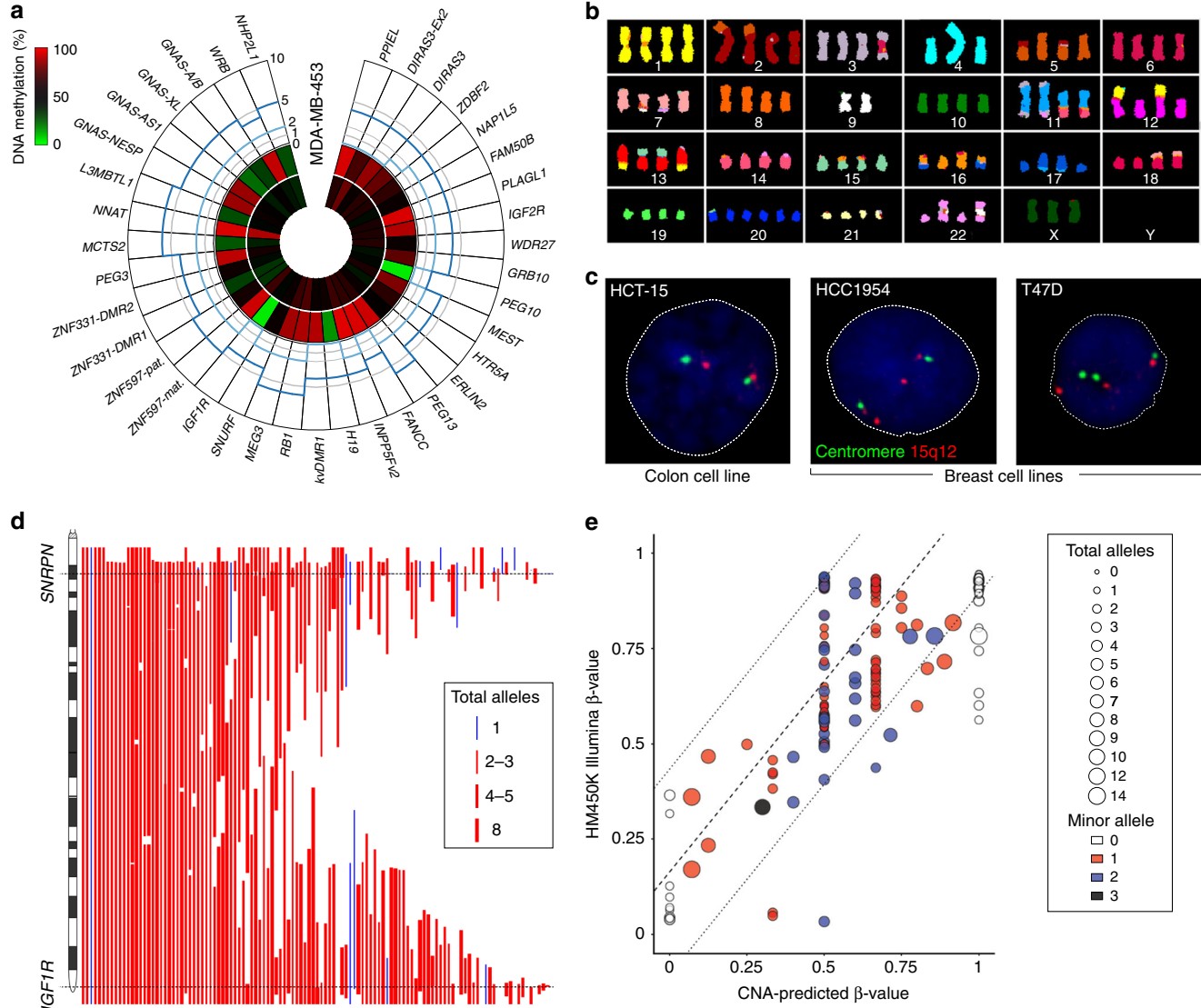

**Fig. 1** Analysis of copy-number and methylation of imprinted regions in cancer cell lines. **a** A Circos graph showing the average methylation for probes mapping within each imprinted DMR for the breast cancer cell line MDA-MB-453. The center heatmap represents the methylation profile obtained from the Illumina Infinium HM450k methylation BeadChip array of 19 normal breast samples, whereas the outer heatmap represent the profile of the cell line. The outer section represents the average copy number for each imprinted loci, obtained from Affymetrics SNP6 array data, depicted as minor allele count (*light blue*) and total copy number (*dark blue*). **b** Sky karyotype validation for the same cell line revealing near tetraploid status. **c** Standard DNA FISH analysis for the chromosome 15q11-13 region in three cell lines. The red probe maps ~1 Mb from *SNRPN* in the *GABRB3* gene, while the green probe maps to the centromere. The colorectal cell line HCT-15 has normal copy-number, whereas the two breast cancer cell lines carry CNAs. The HCC1954 cells have a focal amplification of *SNRPN* while the amplification in the T47D cells incorporates the centromere. **d** A chromosome ideogram showing the extent of amplifications and deletions for the *SNRPN* and *IGF1R* domains on chromosome 15. Amplifications are in *red* and deletion in *blue* with the width of the lines representing the total copy-number for each aberration. **e** The observed vs. expected methylation profile for the *PPIEL* domain in lung-derived cancer cell lines. The dashed lines represent the ± 3 s.d. of the mean of normal control tissues. Data points outside this range are deemed to have a methylation profile independent of CNA

an internal amplification of 15q11.2-12 including *SNRPN* that does not encompass the nearby centromere (Fig. 1c).

**CNAs dictate imprinted DMR methylation profiles**. We hypothesize that CNAs are a major determinant of allelic methylation at imprinted DMRs. To ensure that the profiles obtained using the COSMIC HM450k methylation data set accurately reflected the methylation pattern at imprinted DMRs, we compared the profiles for five cell lines with those obtained using reduced representation bisulphite sequencing (RRBS) generated by ENCODE. Despite the difference in technology, a comparison of the methylation profiles revealed high correlation (Spearman 's correlation: A549 r = 0.85, HCT-116 r = 0.79, T47D r = 0.95, HepG2 r = 0.6, MCF7 r = 0.74.

*P*-value < 0.0001) suggesting that the HM450k methylation array data can be used with high confidence (Supplementary Fig. 7). To test our hypothesis, we subsequently compared the absolute methylation (average methylation for all probes within an imprinted DMR) with the CNA-based predicted methylation value. Assuming that no epimutations have occurred, then methylation will be entirely dictated by chromosome number and parental origin. For example for a normal cell with 2:1 copy-number complement, methylation will be ~ 50% as one allele is methylated and the other unmethylated. However for a cell line of 4:1, depending on the parental origin of the minor allele methylation will be in a ratio of 75:25 or 25:75%. In all cases of cnnLOH the methylation profile will be near zero or fully methylated

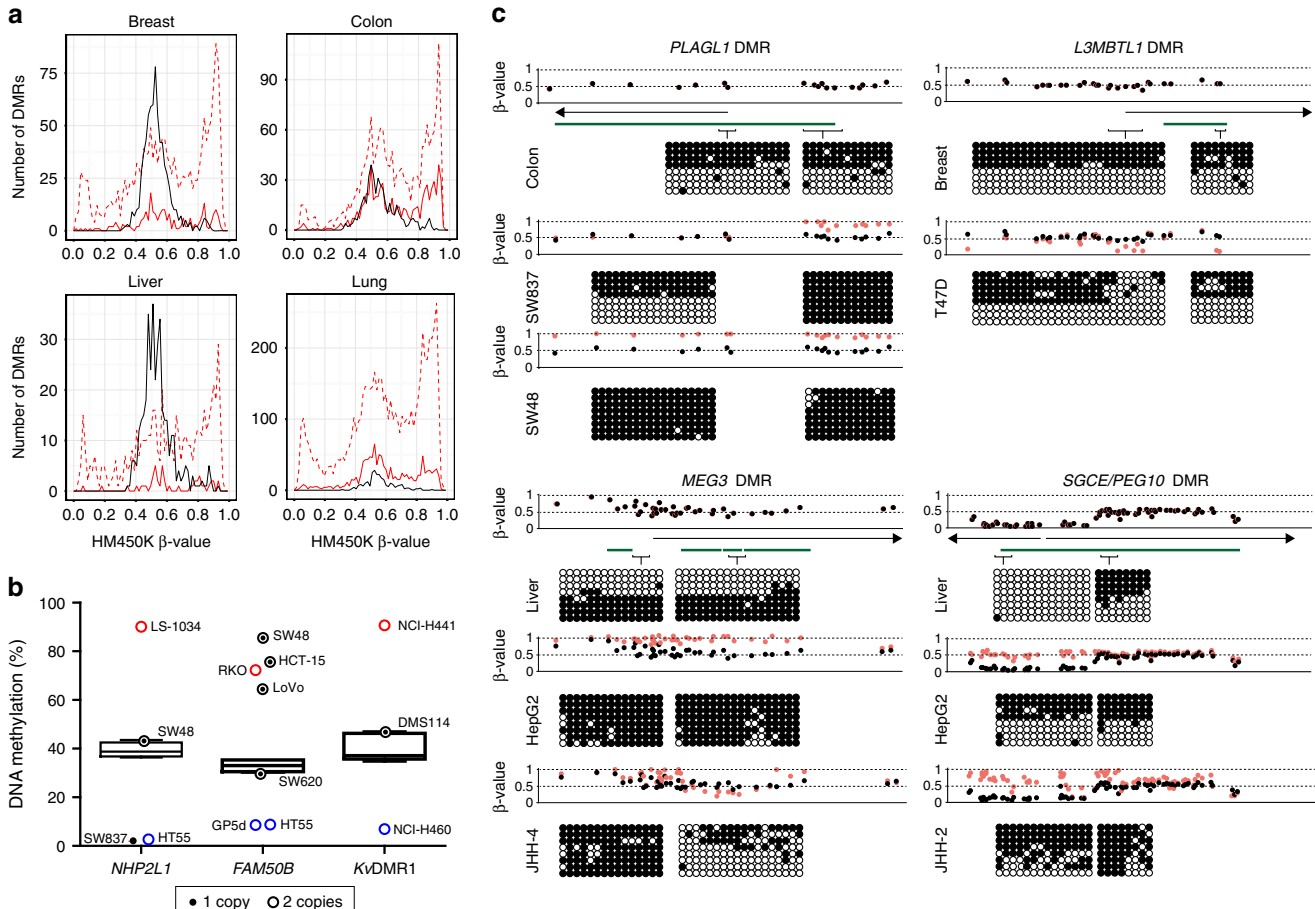

**Fig. 2** Methylation anomalies at imprinted DMRs in cancer cell lines. **a** Line graphs of the number of cancer cell lines showing methylation defects in imprinted DMRs. Control DNA methylation frequencies are depicted by *black solid line* with a β-values near 0.5 indicative of one methylated and one unmethylated allele. *Dashed line* represents total methylation defects irrespective of the underlying CNA status, whereas *solid red line* represents the methylation profiles in samples carrying normal 2:1 copy number complement. **b** Pyrosequencing quantification for the *NH2PL1* and *FAM50B* DMRs in colorectal cell lines with different CNA status. The average methylation of eight controls colon biopsies were used to generate Turkey box-and-whisker plots with whiskers spanning from 25th to 75th percentiles ± 1.5 interquartile range to highlight outliers. A similar analysis was performed for the *KvDMR1* in lung cancer cell lines. Number of copies is represented by *circle size* (*inner*, minor allele; *outer*, total copy number). **c** Detailed methylation maps of four imprinted DMRs as determined by Illumina Infium HM450k methylation beadChip arrays. Each *dot* represent the position of single probes with the profiles of cancer cell lines (*red* data points) compared to the respective normal tissues (*black* data points). The *green bars* highlight the position of the CpG islands associated with each DMR and the *arrow* the nearest transcripts. Two different bisulphite PCRs were performed per region to confirm the strand-specific methylation profile as determined by cloning and direct sequencing. Each *circle* represents a single CpG dinucleotide on a DNA strand (results for multiple DNA strands are depicted as *rows*), *filled circles* indicate a methylated cytosine, and *open circles* an unmethylated cytosine

irrespective of the total chromosome number. Any sample with a CNA-based predicted methylation value outside ± 3 s.d. of normal tissue controls were assumed to have a profile not dependent on CNA, and attributed to a failure to maintain the correct epigenetic profile (Fig. 1e). In total 17.6% of all imprinted DMRs in breast-derived cancer cell lines present with methylation changes independent of CNA, whereas only 12.8% lung, 9% liver and 8.1% colon cancer cell posses methylation profiles not corresponding to copy-number. An estimate of the proportion of methylation variability explained by copy-number alone is shown in the supplementary information (Supplementary Fig. 8). The most frequently epimutated regions varied depending on tissue origin, with the maternally methylated *ZNF597* DMR being affected in 66.6% of hepatocarcinoma cell lines, *H19* in 28.9 % of lung, *ZNF331* DMR1 in 32% of colon and *GRB10* in 44.9% of breast-derived cancer cell lines.

## Methylation profiling at copy-number normal loci. To obtain a clearer picture of the role of aberrant methylation independent of

CNAs, we assessed the allelic methylation asymmetry at imprinted domains in cancer cell lines without copy-number changes. A comparison of the distribution of Infinium HM450k β-values revealed that copy-number normal samples were more frequently hypermethylated rather than failing to maintain methylation, a pattern that was not obvious when all cell lines were analyzed together irrespective of CNAs (Fig. 2a). To confirm the high-density array methylation profiles we performed pyrosequencing for 3 DMRs including *NHP2L1*, *FAM50B* and the *Kv*DMR1 (Fig. 2b). The results obtained were extremely similar for the two different methodologies (Spearman's correlation; *NHP2L1* r = 0.88, *FAM50B* r = 0.97, *Kv*DMR1 r = 0.88), further reassuring us that the array-based methylation profiling is extremely accurate. However, despite being quantitative, pyrosequencing does not give strand-specific or allelic information and averaging β-values for all probes within an imprinted DMR may mask subtle anomalies. For these reasons, we performed bisulphite PCRs and subcloning to confirm some of the rare unexpected array methylation profiles (Fig. 2c). For

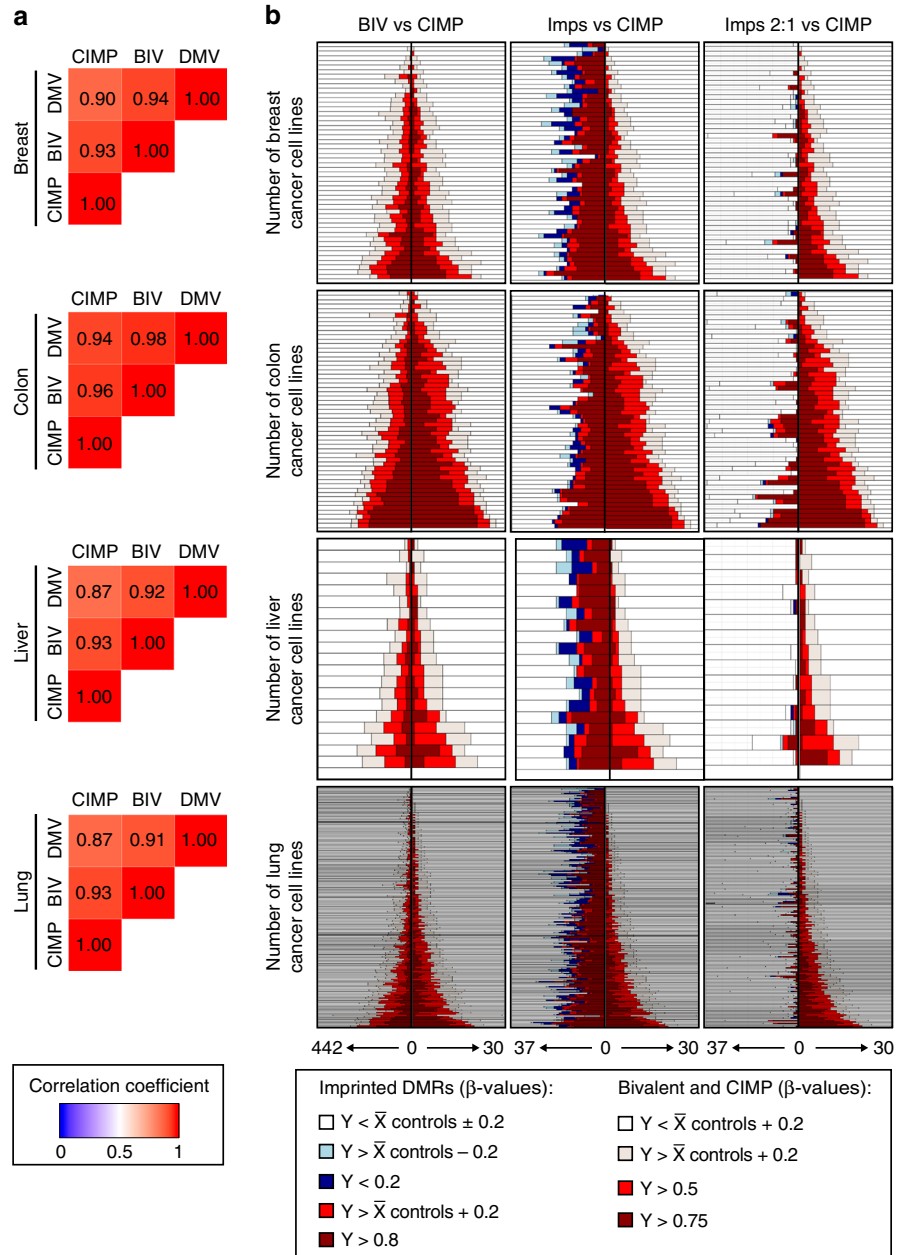

**Fig. 3** The methylation profiles of genomic intervals susceptible to cancer-associated hypermethylation. **a** Heatmap of pairwise correlation coefficient of for CpG island methylator phenotype (CIMP), bivalent domains (BIV) and DNA methylation valleys (DMV) in the cell lines derived from the four different tissues. Numbers in the colored squares represent Pearson's or Sparman's rank according to data distribution. **b** Stacked histograms ranked according to the severity and number of affected loci per cell line. The graphs in the *left column* reveal that cell lines with the highest hypermethylation burden for CIMP regions are similarly hypermethylated at bivalent domains. The *middle column* is a comparison between the methylation profiles of imprinted DMRs irrespective of CNA status and CIMP. The *right row* is the same comparison but with only imprinted domains with a normal copy-number. For each type of loci the number of genes analyzed is indicated on the *x*-axis

example we observe complete hypermethylation at the *PLAGL1* DMR in the colorectal cell line SW48 but two different profiles in SW837, with the 5′ promoter region being fully methylated whilst the interval encompassing the transcription start site (TSS) has normal allelic methylation. A similar bipartite methylation profile was also observed at the *MEG3* DMR in the hepatocarcinoma cell line JHH-4. In addition we observe normal allelic methylation of the *L3MBTL1* DMR in breast cancer cell line T47D, with the exception that the ~ 200 bp immediately adjacent to the TSS, which is devoid of methylation. We previously described that the bidirectional promoter associated with the *SGCE* and *PEG10* is a maternal DMR on the telomeric side of a large CpG island, with

the opposite side unmethylated in all somatic tissues, except placenta where is also methylated on the maternal allele[17]. Intriguingly we observe that several cell lines, irrespectively of tissue origin, adopt this placenta profile, which we confirm by PCR in the hepatocellular carcinoma cell line HepG2 (Fig. 2c).

**The influence of nearby non-imprinted genes within CNAs.** The effect of non-imprinted genes located within the same CNAs as imprinted loci could influence tumor development. In an attempt to understand the impact of additional genes, we identified nine oncogenes and two tumor-suppressor genes

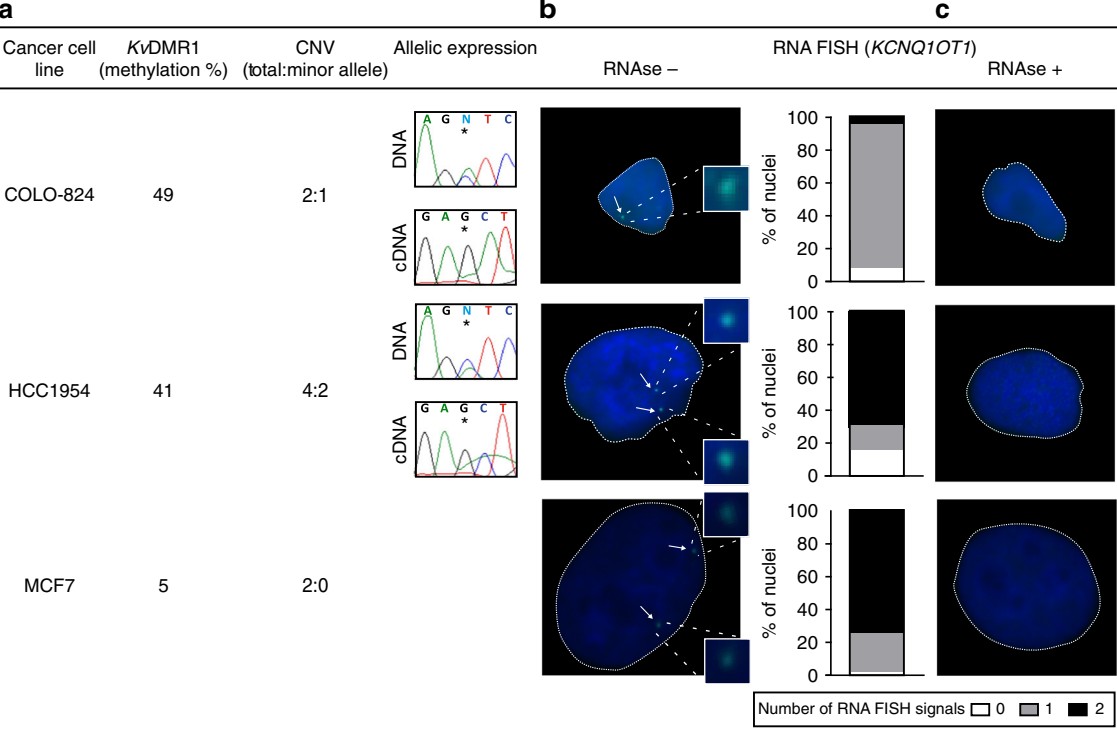

**Fig. 4** Allelic expression analysis of *KCNQ1OT1* in cancer cell lines. **a** Three cell lines with different copy-number status (COLO-824, copy number 2:1; HCC1954, copy number 4:2; MCF7 copy number 2:0) were assessed by Sanger sequencing of rs231359 polymorphism in the *KCNQ1OT1* transcript in DNA and cDNA, respectively. The average methylation of the *KvDMR1* was determined from Infinium HM450k array data. **b** Representative RNA-FISH analysis of *KCNQ1OT1* lncRNA-coated territory (green signal, white arrows) of individual nuclei with inserts representing zoomed in images of FISH signals. Nuclei were stained with DAPI. The quantification of *KCNQ1OT1* expression signals represented as stacked bar charts indicates the percentage of nuclei displaying the indicated number of expression foci. **c** *KCNQ1OT1* RNA-FISH performed on cells retreated with RNAse A

that map within the vicinity of the imprinted loci (Supplementary Data 4). In general, the greater the distance between the cancer-associated gene and the imprinted loci the lower the frequency that both regions are involved in the same cytogenetic aberration. However, in many cases both loci are affected. For example, the imprinted *PPIEL* locus is located ~373 kb from the *MYCL1* oncogene and co-amplification of both regions was observed in 98% of lung, 91% in liver, 97% in breast and 100% in colon cancer cell lines harboring copy-number gains of 1p34.

In addition to the classic acquired LOH caused with deletions, we show that cnnLOH is a common chromosomal defect affecting imprinted loci. Since cnnLOH may lead to homozygosity of pathogenic mutations (i.e., silencing tumor-suppressor genes or activating oncogenes) we screened for genes with homozygous mutations mapping within cnnLOH regions harboring imprinted loci. Of the 280 cell lines analyzed, 258 had at least one region of cnnLOH affecting an imprinted loci with 26% also containing homozygous mutated genes described from in the COSMIC database. The majority of mutated genes have not been associated with tumor initiation of progression. However we did identify reoccurring mutations of *RB1* associated with cnnLOH of chr13q14 and *PTEN* mutations with cnnLOH of chr10q23-26 in lung and breast cancer cell lines (Supplementary Data 5).

Further analysis of the parent-of-origin of the cnnLOH, as inferred by the methylation profile of the affected imprinted DMRs, suggests that the influence on imprinted gene dosage is complex. For example, cnnLOH for the chr11p15.5 interval involves both the maternally and paternally derived chromosomes equally in breast and lung cancer-derived cell lines, but is exclusively paternal in colon cancer cell lines consistent with previous observations that *IGF2* over-expression is oncogenic[7]. For cases also harboring homozygous mutated genes, it seems

that the presence of the genetic variant is more influential than the parent-of-origin of the cnnLOH, with the exception that *RB1*, in which all lung cancer cell lines (7/7) are associated with hypermethylation at the *RB1* imprinted DMR.

**DMR profiles are independent of other epigenome changes.** Compared to normal cells, cancer cells show drastic changes in DNA methylation status, generally exhibiting global hypomethylation at intragenic and repetitive elements accompanied by regions of hypermethylation. Over the years the various regions of cancer-associated hypermethylation have been defined, initially as a collective of promoter CpG islands prone to gaining methylation known as CpG island methylator phenotype (CIMP)[26]. The list of hypermethylated regions is continually expanding with use of technologies with increased genome resolution and now include bivalent domains, CpG-rich sequences decorated with the permissive histone modification H3K4me2/3 and the repressive modification H3K27me3, which represent ~68% of cancer-associated hypermethylated genes[27, 28] and DNA methylation valleys (DMVs), large intervals that are generally devoid of DNA methylation in normal tissues[29] (Supplementary Data 6). We compared the DNA methylation profiles of 30 CIMP, 442 bivalent domains and 166 DMVs to obtain a general genome methylation profile in each cancer cell line. We observe that cell lines with the greatest number of hypermethylated CIMP regions were also more susceptible to bivalent and DMV hypermethylation (Fig. 3a). To determine if the predominant hypermethylation we observe at imprinted DMRs simply reflects the genome-wide methylation state, we ordered each cell line according to its CIMP profile (i.e., ranked according to the severity and number of affected loci per cell line). Unlike CIMP and bivalent domains that are highly

correlated (Pearson's correlation: colon r = 0.96. Spearman's correlation: breast r = 0.93, liver r = 0.92 and lung r = 0.93) there was no apparent similarity between CIMP and imprinted methylation for any cancer type (Fig. 3b).

In addition, we assessed a subset of cancer cell lines for hypomethylation of repetitive DNA (ALU-Yb8, LINE-1 and α-satellites). These elements are commonly used as surrogates to measure genome-wide methylation levels as ~ 27% of total CpG methylation is contained within LINE-1 elements and 23% in ALU-Yb8 sequences[30]. We identified substantial hypomethylation of these elements in numerous cell lines compared to tissue matched controls, but those cell lines affected did not show extreme hyper- or hypomethylation at imprinted DMR (Supplementary Fig. 9). Taken together these observations argue against a direct

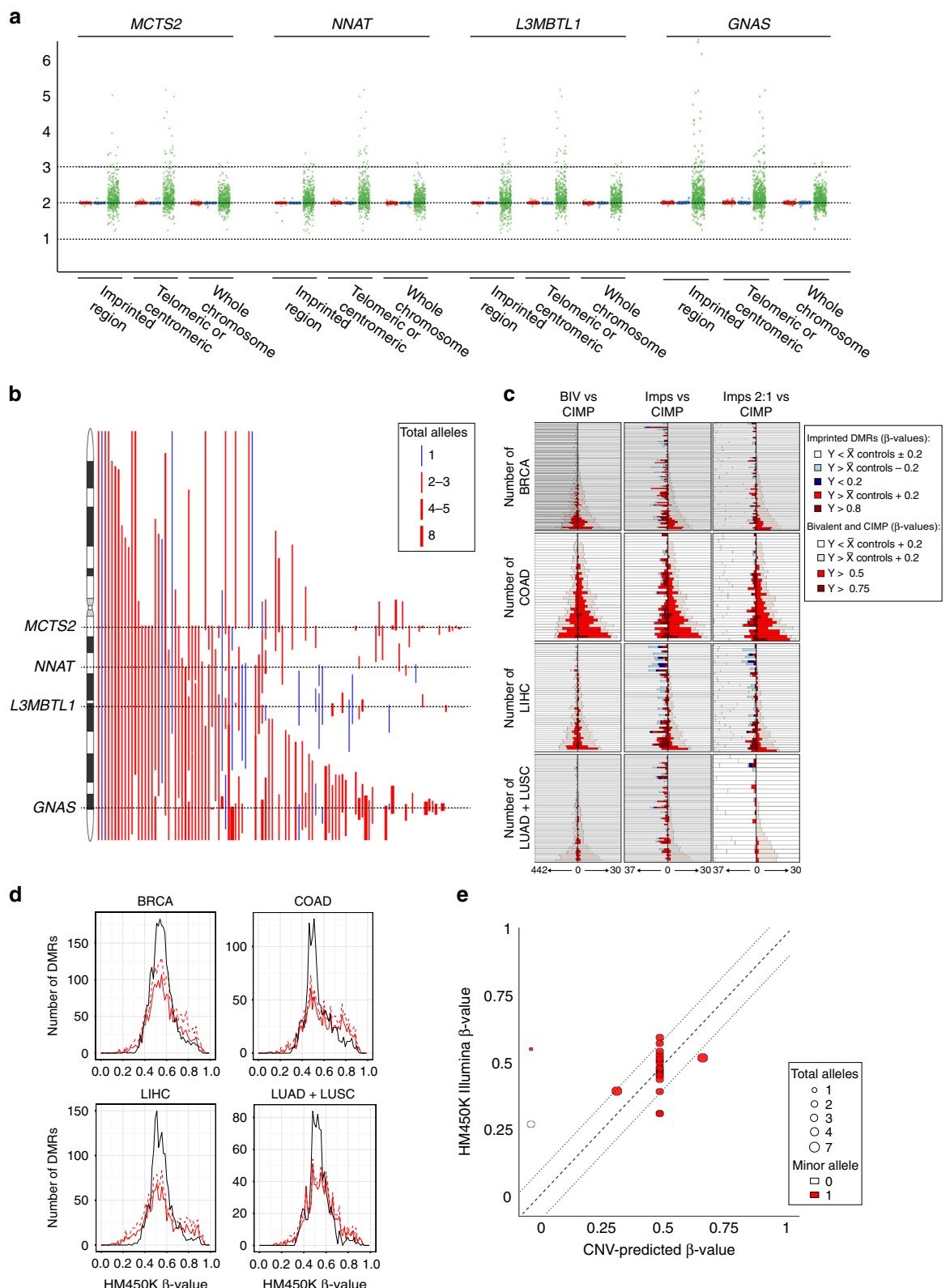

connection between global DNA methylation status and imprinting, supporting an independent mechanism of regulation in cancer.

**Allelic expression profiling in cancer cell lines**. To date, very few cancer cell lines have been assessed for allelic expression of imprinted genes. To determine the influence of both methylation and CNAs on allelic expression, we genotyped highly informative exonic SNPs within *NAPL1L5, PEG10, H19, IGF2, MEST, KCNQ1OT1, MEG3, PEG3, L3MBTL1* and *NHP2L1* in 57 cancer cell lines with allelic expression determined by PCR with reverse transcription (RT-PCR) in heterozygous cases (Supplementary Figs [10]–[19]). This revealed that if expressed, copy-number normal cell lines with normal methylation maintained appropriate monoallelic expression (e.g., *NHP2L1* in NCI-H441; *KCNQ1OT1* in COLO-824, DM-273, HCT-15, NCI-H1048 and MDA-MB-157; *MEG3* in Hs-578-T; *NAP1L5* in BT-474, GP5d and HCT-15; *PEG10* in HCT-116) and in rare cases with robust loss-of-methylation the transcripts were biallelically expressed (e.g., *NHP2L1* in HLE). Interestingly we observe several cases in which cell lines with a total/minor ratio of 4:2 with ~50% methylation are also robustly expressed from one genotype, suggesting that the four chromosomes are maintaining their original parental imprints having two active and two repressed chromosome copies (e.g., *NHP2L1* in Hs-578-T and NCI-H508, *KCNQ1OT1* in AU565 and HCC1954; *H19* in LS-411N and LS-1034; *L3MBTL1* in NCI-H1975; *NAP1L5* in MDA-MB-453). To confirm these expression profiles, we performed RNA FISH for the long non-coding (lnc)RNA *KCNQ1OT1* that is retained near the site of transcription. In the breast cell line COLO-824, we observe a single signal consistent with it being copy-number normal and monoallelically expressed, as did the colorectal cell lines HCT-15, confirming the observations by Sunamura et al.[31] (Fig. [4]a, b and Supplementary Fig. [20]). We observe two signals in the MCF7 cell line that has cnnLOH for 11p15 (total/minor ratio of 2:0) and absence of methylation at the *Kv*DMR1. Interestingly we also observe two *KCNQ1OT1* signals in the copy-number 4:2 HCC1954 cell line consistent with two copies of the paternal allele being expressed in these cells. These observations highlight that fact that a cell may be transcribing from more than one chromosome but maintaining the correct imprinting profile if it carries copy-number aberrations.

**Confirming imprinted CNAs in primary cancer samples**. Human cancer-derived cell lines are widely used models to study the biology of cancer and in the majority of cases recurrent genetic and epigenetic change found in tumors are also identified in cell lines and vice versa. In order to confirm our cell lines observations in primary tumors we analyzed publically available data sets from TCGA including breast carcinomas (BRCA), colon adenocarcinomas (COAD), liver hepatocellular carcinomas (LIHC) and lung tumors (lung adenocarcinomas, LUAD + squamous cell carcinomas, LUSC) for which data is available from normal, tumor and adjacent-tissue samples.

A huge number of CNAs were observed in the primary data sets encompassing imprinted domains, but in most cases copy-number gains were not as severe as in cell lines and were restricted to the tumor samples and not matched adjacent-tissue (Fig. [5]a, b). For many imprinted loci the copy-number aberrations were a mix of both deletions and amplifications, but striking recurrent changes were observed. For example, the four imprinted regions on chromosome 20q (*MCTS2, NNAT, L3MTBL1* and *GNAS*) were invariable subject to copy-number gains with very few deletions observed in the four cancer types. In contrast some imprinted domain were associated with CNAs specific to a tumor type. When observed, *H19/IGF2* CNAs in COAD were mainly amplifications, consistent with previous reports[7], whereas equal numbers of gains/losses are observed in the other three tumor types analyzed. Furthermore, amplifications of *RB1* at chr13q14 were preferentially observed in COAD samples whereas deletions were preferentially observed in BRCA, the latter consistent with the reports of the functional loss of *RB1* in basal-like and luminal breast carcinomas[32]. Similarly chromosome 22q deletions are common somatic alteration in both breast and colorectal cancer[33] and we confirm a high frequency of deletions of *NH2PL1* mapping to 22q13 in BRCA and COAD data sets.

Next, we wish to determine the size of the CNAs since we observed that telomere/centromere bound aberrations tend to be larger than focal internal CNAs in cancer cell lines. This was also true for CNAs encompassing imprinted domains in primary cancer samples, with focal internal CNAs being more prevalent and often in higher copy number than those involving either the telomere or centromere (extended data available at the www.humanimprints.net). For example the average telomere bound amplification for *IGF1R* on chromosome 15 is 18 Mb, often representing just a single additional copy, whereas internal amplifications are on average 3 Mb with 6–7 copies.

**Imprinted methylation in TCGA data sets**. The methylation changes in the TCGA primary tumor samples are less frequent and less extreme than in cancer cell lines; however, this may reflect cellular composition of the sample with primary tumors being heterogenous and infiltrated with normal cells (mean tumor cell estimate $\pm$ s.d.; BRCA 72.2 $\pm$ 20.5%, COAD 71.5 $\pm$ 16.6%, LIHC 87.1 $\pm$ 11.7%, LUAD 76.3 $\pm$ 19.1%) (Supplementary Data [7]).

There was high correlation between hypermethyalted CIMP, bivalent domains and DMVs for all tissues (Spearman's correlation for CIMP vs bivalent domains: BRCA r = 0.88, COAD r = 0.95, LIHC r = 0.81 and LUAD r = 0.84)(Fig. [5]c), with the

**Fig. 5** Methylation profiles of imprinted DMRs in primary TCGA cancer samples. **a** Scatter plots for imprinted domains on chromosome 20 in tumor, adjacent and control somatic DNA reveals that the CNAs are restricted to the cancer samples. **b** Ideogram for chromosome 20 showing the extent of amplifications and deletions for *MCTS2, NNAT, L3MBTL1* and *GNAS* domains in lung-derived cancer samples. Amplifications are in *red* and deletion in *blue* with the width of the lines representing the total copy-number for each aberration. **c** Stacked histograms ranked according to the severity and number of affected loci per sample. The graphs in the *left column* reveal that cell lines with the highest hypermethylation burden for CIMP regions are similarly hypermethylated at bivalent domains. The *middle column* is a comparison between the methylation profiles of imprinted DMRs irrespective of CNA status and CIMP. The *right row* is the same comparison but with only imprinted domains with a normal copy-number. For each type of loci the number of genes analyzed is indicated on the *x*-axis. **d** Line graph of the number of cancer samples showing methylation defects in imprinted DMRs. Control DNA methylation frequencies are depicted by *black solid line* with a β-value near 0.5 indicative of one methylated and one unmethylated allele. *Dashed red line* represents total methylation defects irrespective of the underlying CNA status, whereas *solid red line* represents the methylation profiles in samples carrying normal 2:1 copy number complement. **e** The observed vs. expected methylation profile for the *GNAS* A/B DMR overlapping exon 1A in lung tumors. The *dashed lines* represent the $\pm$3 s.d. of the mean of normal control tissues. Data points outside this range are deemed to have a methylation profile independent of CNA

COAD sample set being more hypermethyalted than the other cancer types (Supplementary Data 8). With the exception of the LIHC samples, the number of imprinted DMRs subject to hypermethylation is much greater than those with hypomethylation (Fig. 5c, d), with no correlation with global DNA methylation status as gauged by comparisons with CIMP, bivalent domains and DMVs (Fig. 5c). Consistent with our observation in cell lines, aberrant imprinted methylation was closely associated with CNAs status (Fig. 5e). To determine whether specific imprinted DMRs were affected in cancer from different tissue origin, we assessed the frequency of methylation defects at loci with normal 2:1 copy-number. No particular DMR was affected in specific cancer type. The most frequently observed hypermethylated regions are PEG13 (60%) in COAD, PPIEL (34%) in LIHC, INPP5FV2 (30%) in BRCA and the paternally methylated ZNF597 DMR (20%) in LUAD. Similarly, no imprinted DMR was consistently hypomethylated, with the most frequently affected regions being GNAS (20%) in LIHC, FAM50B (9%) in LUAD, L3MBTL1 (8%) in COAD and MCTS2 (6%) in BRCA. Furthermore some imprinted DMRs have stable methylation comparable with normal tissues with the LUAD data set having the most unaffected loci (NAP1L5, PLAGL1, PEG10, MEST, PEG3, NNAT and NH2PL1).

**Molecular subtypes and imprinted methylation**. The somatic load of driver mutations in the cancer cell lines has been extensive studied, with the results cataloged in the COSMIC somatic mutation database. To determine if methylation defects at imprinted loci were more prevalent in tumors harboring-specific genetic aberrations, we compare methylation profiles of samples with 2:1 copy-number positive for common mutations[1]. We failed to observe any association between reoccurring mutations (TP53, PIK3CA, CDH1 and AKT1 in breast-derive cancer cell lines; TP53, MLH1, APC, KRAS, BRAF and PIK3CA in colorectal cancer cell lines; CTNNB1, TP53, AXIN1, JAK1 and LRP1B in hepatocarcinoma cell lines; TP53, PIK3CA and AKT1 in lung cancer cell lines) or microsatellite instability (MSI) and imprinted DMR status for any cell lines. However we observe increased normal copy-number status in colon, liver and lung-derived cell lines with high MSI, consistent with the previous reports that chromosomal instability and microsatellite stability are largely mutually exclusive[34] (Supplementary Figs 21–24).

The primary tumor samples from the TCGA have also been extensive studied at the molecular level allowing similar comparisons to be performed. Comparable to cell lines, we do not observe increased frequency of aberrant imprinted methylation for any characteristic including reoccurring mutations, hepatitis infections and alcohol consumption in LIHC, smoking status in LUAD or HER2, progesterone or estrogen hormone receptor status in BRCA data sets (Supplementary Figs 25–28).

## Discussion

In this current study, we have generated DNA methylation and CNAs profiles for over 280 cancer cell lines and compared the results with large primary tumor data sets available from the TCGA.

Methylation at imprinted loci is unique in that opposing parental alleles have different methylations states that are dictated by parental origin. Unlike other single-copy regions which may be prone to accumulate cancer associated hypermethylation from an initially unmethylated state, imprinted DMRs can either loose or gain methylation. Importantly cancer-associated copy number changes will influence the absolute methylation observed at imprinted DMRs while this is not so important for CIMP, bivalent domains and DMVs. In the majority of cases, we observe

that the methylation profiles at imprinted DMRs do not represent epigenetic changes but simply the parental origin of underlying CNAs. In some cases, we predict methylation errors may have occurred within cytogenetically abnormal genomes, but it is not possible to know which aberration, the CNA or methylation defect, occurred first. In addition, it is impossible to know if all copies of a highly amplified region for example PPIEL or GNAS are subject to epimutations, resulting in a mosaic/clonal state.

It is known that some tumors with higher frequency of cancer-associated hypermethylation are enriched for oncogenic mutations. However our detailed analysis was unsuccessful in revealing a clear link with any driver mutation or exposure. Mutations in BRAF[V600E][35] and CTNNB1[36] have been described in colorectal and liver tumors with extensively acquired hypermethylation. We fail to identify any association between aberrant imprinted methylation and these two mutations, despite one study in hepatocellular carcinomas reporting enrichment of CTNNB1 mutations in samples with hypomethylation at imprinted loci[37].

Direct evidence for the role of LOI and cancer comes from the mouse model, in which biallelic expression of Igf2 present on an Apc(Min) background has an increased incidence of colorectal cancers[38]. However, human data implicating this locus in sporadic cancer is more controversial. Biallelic expression of IGF2 has been associated with hypomethylation of the H19 DMR in colorectal cancer[39], rather than the hypermethylation that would predictably lead to reactivation of the maternal allele of IGF2 by ablation of insulator function. This hypomethylation extends to the IGF2-DMR0 region and was reported to occur in the normal colonic mucosa of patients, suggesting that this epigenetic abnormality is not limited to the tumors. Methylation data from TCGA COAD data set supports the observed hypomethylation at both these paternally methylated DMRs (cg19642877 is the only probe on the HM450k platform mapping within IGF2-DMR0 with methylation somatically acquired DMR, therefore this was not included in our analysis) but there is no evidence for aberrant methylation in adjacent normal tissues (Supplementary Data 9), endorsing the reports that methylation at this regions is not a predisposing constitution biomarker for colorectal cancer[40].

Cultured cancer cell lines are widely used models and have help form the fundamental basis for our current understanding of cancer biology. However the use of such an in vitro system has limitations. These include cell line misidentification[41] and inter-laboratory clonal expansion or induce secondary genomic changes that may occur since these originally heterogeneous cell lines have been maintained for decades in different culture conditions. To limit these oversights, we have performed our genotyping and methylation analyses on the same DNA samples obtained from the Wellcome Trust Sanger Institute, with quantitative and allelic conformation performed on these DNA aliquots. Furthermore the results of our allelic expression profiling are consistent with examples reported in literature. For example, the methylation and allelic expression of DIRAS3 in the breast cell lines UM159 and CAL51 as reported by Niemczyk et al.[42], and the H19 methylation profile of the colorectal cell lines COLO-205 and LoVo are similar to those reported more than 10 years ago[43]. We also observe identical hypermethylation in the liver cancer cell lines at the RB1 (HLE and HuH-7) and MEG3 DMRs (HLE and HuH-7)[44–46] although our data suggests that these profiles are due to cnnLOH.

While we are confident that our allelic expression results are consistent with previous reports and reflect the expression status in the cell lines, results are sometimes not as would be predicted. For example in the colon cell line HCT-116, we observe robust monoallelic expression despite the H19 DMR being methylated on both alleles. This is further complicated by the fact that

the nearby *cis*-regulated somatic *IGF2*-DMR0 maintains allelic methylation consistent with previous reports[39]. In addition the COLO-205 is heterozygous for the reciprocally imprinted, co-regulated *H19* and *IGF2* genes. However, we observe a discordant expression pattern with maintained monoallelic expression of *H19* and biallelic expression of *IGF2*, a situation that has previous been reported in lung adenocarcinomas[47]. One possible explanation for the latter observation is that expression of *IGF2* has switched from the predominant P3/P4 imprinted promoters to the tissue-specific non-imprinted P1 promoter[48]. A similar scenario may account for the biallelic expression of *MEST* observed in many cell lines despite the DMR being hypermethylated, however since the SNP used to discriminate alleles is located beyond PCR range, we are unable to demonstrate promoter switching.

To conclude, the involvement of imprinting in cancer has gained much attention, but very few studies have taken CNAs into account when reporting methylation profiles. This has resulted in the inflated documenting of somatically acquired epigenetic errors at imprinted loci in cancer. Our data strongly suggest that CNAs should be investigated before reporting an epigenetic change at imprinted loci in cancer.

## Methods

**DNA samples.** Cell line DNA used for confirmation analyses were obtained directly from the Sanger Institute. All cell lines cultured in-house, except two, were obtained from ATCC repository and grown using conditions recommended by the supplier. The hepatocellular carcinoma cell lines JHH-2 and JHH-4[49] were obtained from the National Institute of Biomedical Innovation JCRB Cell Bank, Osaka, Japan. These cells were maintained in Eagle's minimal essential medium plus 10% fetal bovine serum.

Control liver, breast, colon and lung tissue samples were obtained from Catalan Tumor Biobank (http://www.clinicbiobanc.org/) with DNA extracted with phenol/chloroform using phase-lock gel columns (5 Prime) and RNA extracted with Trizol (Thermofisher) following standard protocols. cDNA synthesis was performed using 1ug of RNA following an initial DNase treatment (amplification grade DNase I, Invitrogen). Reverse transcription was performed with MMLV retrotranscriptase (Promega) and random primers (Promega) following the manufacturer's instructions. Ethical approval for this study was granted by the Institutional Review Boards of the Bellvitge Institute for Biomedical Research (PR223/09) and (PR024/11).

**Bioinformatic analysis.** The Illumina Infinium HM450k methylation data sets for cancer cell lines and primary samples was obtained from COSMIC cell line project (GEO number GSE68379) and TCGA Data Portal (https://tcga-data.nci.nih.gov), respectively. Pre-processing and normalization were performed following GenomeStudio (Illumina) using the Bioconductor minfi package[50]. This includes background subtraction and control normalization. After normalization, methylation beta (β) values range from 0 (unmethylated) to 1 (fully methylated). Before analysis, we excluded possible sources of technical bias and genetic variation that could influence results. We discarded, firstly, all probes with a detection *P*-value > 0.01 in more than 10 % of the samples and secondly, those with missing values in one or more of the tested samples. We also discarded those probes containing a common SNP (minor allele frequency ≥ 1%, dbSNP138) in the flanking 5 bp of the interrogated CpG site. Finally, we removed all probes mapping to the X and Y chromosomes. All analyses were performed using human genome version 19 (hg19) as the reference genome. When applicable, coordinates from hg18 to hg19 version were converted using LiftOver tool from USCS Genome Browser website (https://genome.ucsc.edu).

Epigenetic profiles: Infinium HM450k probes located within bivalent domain, DMV and CIMP regions were extracted according to published coordinates[26, 27, 29]. For each sample the average β-value of the probes located within each defined interval was calculated. For subsequent methylation analysis, only regions that were uniformly unmethylated (<0.2 average β-value) in normal tissues (*n* = 19 breast, *n* = 12 colon, *n* = 10 liver and *n* = 7 lung) were selected. We categorized the intervals according to the severity of methylation gains as compared to their corresponding normal tissue controls, being lowly methylated (mean of controls + 0.2), mildly methylated (absolute β-value > 0.5) and highly methylated (absolute β-value > 0.75).

Imprinted methylation analysis: Infinium HM450k probes located within 37 germline imprinted DMRs which have more than one probe were extracted from loci defined by methyl-seq[17]. For each sample, the average methylation at each imprinted DMRs was calculated. Briefly, we classified hyper/hypomethylated imprinted DMR when averaged β-values are above or below 0.8/0.2. We also calculated intermediate methylation defects when samples show ± 0.2 β-value relative to controls, but the absolute methylation do not reach 0.8/0.2 cutoff values.

For each cancer type, methylation analyses for cell lines and TCGA samples were performed independently using the same criteria. For TCGA data set, methylation level of an individual tumor sample was compared to its corresponding non-tumor adjacent-paired tissue, whereas methylation level for each cancer cell line was compared to the averaged β-values of corresponding normal tissue controls.

**RRBS methylation analysis.** Processed-RRBS data for A549, T47D, MCF7, HCT-116 and HepG2 cell lines were downloaded from ENCODE project[51] (GEO accession number: GSE27584). At each CpG site, the methylation rate was calculated as the ratio of methylated reads (C) over the total read count (C + T) at that site. Similar to Infinium HM450k data, CpG sites within imprinted DMRs were extracted according to coordinates[17] and the average methylation at each imprinted DMR was calculated using RRBS-detected cytosines covered by ≥ 10 reads. The association between methylation level at imprinted DMRs obtained from Infinium HM450k and RRBS methods was assessed using Spearman's correlation coefficient.

Predicted methylation analysis: we obtained Affymetrix Genome-wide SNP6.0 genotyping profile of 287 cancer cell lines and 2676 TCGA primary tissues from the Sanger Institutes COSMIC cell line project and TCGA data portal. A total of 306 cancer cell lines (derived from 51 breast cancers, 54 colorectal cancers; 20 hepatocarcinomas and 181 lung tumors) had Infinium HM450k array data, allowing us to determine the association between CNAs and DNA methylation. At the time of this study (2013 onwards) only 178 TCGA primary tissues had pair data copy-number and methylation data. To obtain the CNA predicted methylation value we first determined the minor methylation value as the ratio between the minor and the total allele count. Then, if this corresponded to an Infinium HM450k value below 0.5, we assume that the methylated allele was the minor one and reported the minor methylation value as the CNA predicted methylation. If the corresponding Infinium HM450k value was above 0.5, we assumed that the methylated allele was the major one and report 1—minor methylation value as the CNA predicted methylation.

To estimate the proportion of methylation variance explained by copy-number we used univariate linear regression. For each imprinted region and tissue type, we fit a model with the average observed methylation as outcome and the expected-by-copy-number methylation (described above) as predictor, considering the different cell lines as observations and obtaining the corresponding coefficient of determination (R²).

CNA analysis: We used Level 3 CNA estimations for TCGA samples (BRCA *n* = 1044; COAD *n* = 429; LIHC *n* = 208; LUAD *n* = 494 and LUSC *n* = 501). For the cell lines, we used the CONAN copy number estimations. CNAs were selected if they overlapped an imprinted DMR by at least 1 bp. We defined an amplification/deletion to be internal if it was more than 1 Mb away from the corresponding telomere/centromere (whichever is closer). For TCGA data, we utilized all available data (even if no methylation data was available) and estimated copy number using circular binary segmentation algorithm as implemented in PSCBS[52].

Mutation status in cancer cell lines: We obtained mutational data for all cell lines from the repository Cell Line Project release v76 (http://cancer.sanger.ac.uk/cell_lines).

**Genotyping and imprinting analysis.** Exonic SNPs (minor allele frequency > 0.1) within imprinted transcripts were identified in the UCSC hg19 browser and genotypes obtained by PCR and direct sequencing. Sequence traces were interrogated using Sequencher v4.6 (Gene Codes Corporation, MI). Heterozygous sample were analyzed for allelic expression using RT-PCR incorporating the polymorphism within the final PCR amplicon (for primer sequences see Supplementary Data 10).

**Bisulfite PCR of cell line-derived DNA.** For confirmation of methylation profiles in normal tissues and cancer cell lines standard bisulfite conversion of approximately 1 μg DNA was subjected to sodium bisulphite treatment and purified using the EZ DNA methylation-Gold kit (ZYMO, Orange, CA). Approximately 2 ul of converted DNA was used in each amplification reaction using Immolase Taq polymerase (Bioline) at 45 cycles and the resulting PCR product cloned into pGEM-T easy vector (Promega) for subsequent subcloning and sequencing, (for primer sequence see Supplementary Data 10).

**Pyrosequencing analysis for methylation quantification.** Approximately 50 ng of bisulfite converted DNA was used for pyrosequencing of imprinted DMR or repeat elements[30, 53]. Standard bisulfite PCR was used to amplify the regions of interest with the exception that one primer was biotinylated (for primer sequences see Supplementary Data 10). For sequencing, forward primers were designed to the complementary strand. The pyrosequencing reaction was carried out on a PyroMark Q96 instrument. The peak heights were determined using Pyro Q-CpG1.0.9 software (Biotage). The data were subjected to statistical analysis using Graph Pas Prism 5.0 (Graph Pad Software). Difference between groups were considered significant at *P* < 0.05.

**DNA FISH.** The detection of CNAs involving 15q11-13 was identified using Vysis probes with *GABRB3* labeled with SpectrumOrange and the D15Z1 CEP15 labeled

with SpectrumGreen (Abbott Molecular Inc.). For the preparation of metaphase spreads, cell lines were exposed to colcemid for 3 h ~24 h after feeding. After treatment with hypotonic solution the cells were fixed in Carnoy's fixative dropped onto glass slides. Prior to hybridization the slides were dehydrated through an ethanol series of 70, 85 and 100% for one minute each. Heat denaturation, hybridization in humidified chambers and washes were performed following supplier's recommendations. DAPI-vectorshield mounted slides were visualized with on a Zeiss microscope with images of individual nuclei captured using the automated MetaSystems platform.

**RNA FISH**. Expression of the lncRNA *KCNQ1OT1* assessed by RNA FISH. Cells were grown on poly-L-lysine coated slides and fixed in 4% paraformaldehyde-PBS for 20 min at room temperature, washed twice in PBS and permeabilized with 70% EtOH at 4 °C overnight. The fixed slides were washed twice in PBS, followed by two washes in 2× SSC. Control slides were treated with RNAse A (Roche) treatment for 1 h at 37 °C and subsequently washed twice in 2× SSC. A pre-hybridization step involving incubating the fixed slides in 15% formamide/2× SSC for 20 min at room temperature was performed before hybridization. BAC probe RP11-937O11 (obtained from the BACPAC Resource Center) was labeled with either SpectrumRed or SpectrumGreen UTP by nick translation and competed with human COT-1 DNA (Roche) following the supplier's protocol (Abbott Molecular Inc.). 4 ul of heat denatured probe/LSI buffer was added to non-denatured slides and hybridized overnight at 37 °C in a humidified chamber. Subsequently the slides were carefully washed in 2× SSC/15% formamide for 20 min at 37 °C, followed by two washes in 1× SSC at 37 °C for 20 min each. Slides were mounted using DAPI-vectorshield and visualized using a Leica fluorescence microscope. A total of 100 nuclei were counted to determine the number of expression foci per cell.

**Data availability**. The Illumina Infinium HM450k methylation array and RRBS methylation data sets for cancer cell lines were obtained from Gene Expression Omnibus (GEO) accession numbers GSE68379 and GSE27584, respectively. Circus graphs for methylation and copy-number as well as maps for the extent of amplifications and deletions for all cell lines is available at www.humanimprints. net.

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

## Acknowledgements

We are grateful to Paul Edwards at the Hutchison-MRC Research Centre, Cambridge for helpful discussions and sharing of Sky karyotype data. This work was supported by Spanish Ministerio de Educación y Competitividad (MINECO) (BFU2011-27658 and BFU2014-53093-R to D.M.), co-financed by the European Regional Development Fund (FEDER), the Asociaciòn Española Contra el Cáncer (AECC to D.M.) and the Fundaciò Olga Torres (FOT to H.H.). AMS is a recipient of a FPI PhD studentship from MINECO (BFU2014-53093) and H.H. is a Miguel Servet (CP14/00229) researcher funded by the Spanish Institute of Health Carlos III (ISCIII). Work supported by the Xarxa de Bancs de Tumors de Catalunya is sponsored by Pla Director d'Oncologia de Catalunya (XBTC).

## Author contributions

D.M., A.M.-T. and E.V. designed the experiments and analyzed the data. A.M.T., A.M.-S., M.S.S. and S.M. performed the experiments and contributed to discussion. D.M. wrote the manuscript. A.M.T. and M.S.D. prepared the figures. J.R.H.M., H.H., M.G. and M.E. contributed to data analysis and discussion. All authors revised and edited the manuscript.

## Additional information

**Competing interests:** The authors declare no competing financial interests.

