## [Peer Review File · Nature Communications]

Reviewers' comments:

Reviewer #1 (Remarks to the Author):

The authors have sought to address an important question in, interaction and interplay between methylation, copy number and gene expression. This is a very important question, particularly in the cancer setting and one which as the authors highlight is generally not considered. The authors show quite nicely the influence of copy number alterations and aberrant methylation on gene expression at imprinted regions. Furthermore gaining an understanding of how imprinted regions are affected will untimely allow inferences to be made about the rest of the cancer genome. Overall I have few comments:-

- The authors have focused exclusively on cancer cell lines and primary tissues primarily from the TCGA. Although this has provided excellent data, it would be interesting to see the interrogation of the combined bisulphite sequencing, expression and copy number data from ENCODE (or those generated by iHEC) which may also be used to further investigate the effect of copy number on methylation.
- The authors perform selected pyro sequencing and targeted bisulphite sequencing. Although these data appear to support their results, it would be nice to have a comparison across all regions. Are there any overlapping ENCODE samples with RRBS or WGBS which would allow all imprinted loci to be assessed, even if just across a small cohort. Or alternately they could compare publically available RRBS/WGBS and copy number data from primary cancers to the data from the TCGA.
- Is it possible to work out the cellular timing of alterations?
- The interaction of methylation and copy number of gene expression is a key question in understanding the functional effects of these perturbations on cancer development. Have the authors attempted to look outside imprinted genes, particularly were non-imprinted genes sit within the same genomic alteration, and as such can they start to tweeze out the relationship between epigenetic and genetic alterations in such regions?

Reviewer #2 (Remarks to the Author):

This paper brings together large datasets from the COSMIC and TCGA and reaches the conclusion that the majority of methylation abnormalities at imprinted gene loci occurring in cancer cell lines and primary tumors result from copy number abnormalities and not epimutations. The conclusions drawn by the authors are of interest and the availability of the data at www.humanimprints.net will benefit the scientific community. However I would suggest that the abstract and title should specify that the analysis was for four human cancer types (lung, colorectal, breast and hepatic cancers).

It would be helpful if the authors could clarify at which imprinted loci there were no apparent relationship between CNAs and the parent-of-origin of the amplified/deleted chromosome and which loci (and tumor types) there appeared to be a relationship.

There are some spelling mistakes and in some the grammar required correcting

Reviewer #3 (Remarks to the Author):

In this manuscript the authors analyze imprinted differentially methylated regions (DMRs) across cancer cell lines and TCGA data. They claim that in the majority of cases methylation at DMRs reflect the underlying copy number of the region. While this result is not surprising, as far as I know, it has never been published before and the author's data may be a useful resource to scientists studying methylation in tumours.

Specific comments.

Imprinted loci that are amplified to high levels maybe those that are close to focally amplified oncogenes. For example PPIEL is ~300 kb from the oncogene MYCL1. The authors should comment on the frequency of these occurrences.

Figure 1A – It would be helpful if the loci names were appended with chromosomal arm locations so that you could compare this data to the Sky karyotype in figure 1B. The labels of copy number to the left of MDA-MB453 are too small to be seen.

Is the SNURF locus in Figure 1A the same as the SNRPN locus in Figure 1D? If so the same name should be used throughout the manuscript.

The authors claim that for chromosomes harboring more than one imprinted domain, each locus is associated with private focal CNAs. However in figure 1D, it appears that in ~25% of cell lines the SNRPN and IGF1R loci are co-amplified due to a chromosomal arm level gain of 15q. These amplifications are neither private nor focal CNAs.

Copy number is not interpretable without understanding the baseline ploidy of tumours. For instance a segment with a copy number of 3 is an amplification in a diploid tumour but a deletion in a tetraploid tumour that has undergone whole genome doubling (MDA-MB-453 is an example of this). The authors use of copy ratios shows a good understanding of this issue. But if tumours shown in figure 1D are of different baseline ploidy than copy number should be shown relative to the baseline ploidy (total alleles/baseline ploidy). Copy number shown in tables also should include baseline ploidy or relative copy number.

Copy number changes shown in tables should also give some indication on whether these changes are due to focal or arm level alterations. One way of doing this is to indicate what % of the total chromosomal arm do these alterations represent.

Figure 2C is difficult to understand and should have a legend. It is unclear what the black and white circles underneath methylation maps of loci are suppose to represent.

In figure 3B, it is unclear what exactly the X axes of the graphs are.

In figure 4B the RNA FISH signals for cell line MCF7 are barely visible. Can you show a better representative cell for this figure?

For figure 5B the same comment as figure 1D. If the data in 5B is not relative copy number than how did the authors take in account differences in tumour purity when calculating copy number?

For figure 5c the same comment as for figure 2C.

REVIEWER 1.

We thank the reviewer for their positive comments about our work.

Question 1.

The authors have focused exclusively on cancer cell lines and primary tissues primarily from the TCGA. Although this has provided excellent data, it would be interesting to see the interrogation of the combined bisulphite sequencing, expression and copy number data from ENCODE (or those generated by iHEC) which may also be used to further investigate the effect of copy number on methylation.

Answer 1.

In order to compare the imprinted methylation profiles for cancer cell lines utilizing the COSMIC datasets with those available from ENCODE, we interrogated the reduced representation bisulphite sequencing (RRBS) datasets for five cell line which were common in both collections (MCF7, HepG2, T47D, HCT116 and A549). Unfortunately copy-number data is not available at ENCODE for these samples. The ENCODE methylation data allowed for the examination of 23-29 of the 37 imprinted DMR analysed by HM450k array (with CpG read depth >10 for 2 replicates). Despite the difference in technology a comparison of the methylation profiles revealed high correlation (Spearman r , A549 $r = 0.85$; HCT116 $r = 0.79$; T47D $r = 0.95$; HepG2 $r = 0.6$; MCF7 $r = 0.74$. $p < 0.0001$) suggesting that the HM450k methylation data described in this manuscript can be used with high confidence.

Graphs showing the correlations for methylation levels at imprinted loci as determined by RRBS ENCODE and HM450k methylation array from COSMIC.

Question 2.

The authors perform selected pyro sequencing and targeted bisulphite sequencing. Although these data appear to support their results, it would be nice to have a comparison across all regions. Are there any overlapping ENCODE samples with RRBS or WGBS which would allow all imprinted loci to be assessed, even if just

across a small cohort. Or alternately they could compare publically available RRBS/WGBS and copy number data from primary cancers to the data from the TCGA.

Answer 2.

We tried to utilize ENCODE WGBS datasets but the data was not of sufficient quality to allow comparisons with the HM450k methylation array data. To allow for comparisons across additional imprinted regions we performed an extended pyrosequencing analysis for more than 20 imprinted DMRs in four cell lines (MCF7, HepG2, T47D and HCT116). This comparison also revealed high correlation (Spearman r , HCT116 $r = 0.58$; T47D $r = 0.79$; HepG2 $r = 0.74$; MCF7 $r = 0.5$. $p < 0.01$).

Graphs showing the correlations for methylation levels at imprinted loci as determined by pyrosequencing and HM450k methylation array from COSMIC.

A sentence has been added to the manuscript describing the comparisons between different technologies to quantify methylation and the correlations obtained. It reads "To ensure that the profiles obtained using the COSMIC HM450k methylation dataset accurately reflected the methylation pattern at imprinted DMRs, we compared the profiles for five cell lines with those obtained using reduced representation bisulphite sequencing (RRBS) generated by ENCODE. Despite the difference in technology, a comparison of the methylation profiles revealed high correlation (Spearman r , A549 $r = 0.85$; HCT116 $r = 0.78$; T47D $r = 0.95$; HepG2 $r = 0.6$; MCF7 $r = 0.77$. $p < 0.0001$) suggesting that the HM450k methylation data can be used with high confidence".

In addition we have included a brief description of the analysis in the methods section.

Question 3.

Is it possible to work out the cellular timing of alterations?

Answer 3.

In an attempt to determine the cellular timing of the alterations we examined the HM450k methylation profiles in five patient-derived tumour xenograph models for breast cancers. In each case we examined the profiles of the 37 imprinted DMRs in early and late passage PDTXs but we failed to identify any methylation or CNAs changes. This suggests that cancer-associated aberrations had already occurred in the primary tumour and do not continually evolve in this model system.

Heatmap of the HM450k probes located within known imprinted DMRs in control breast tissues (n=19) and the five PDTX models.

Question 4.

The interaction of methylation and copy number of gene expression is a key question in understanding the functional effects of these perturbations on cancer development. Have the authors attempted to look outside imprinted genes, particularly were non-imprinted genes sit within the same genomic alteration, and as such can they start to tweeze out the relationship between epigenetic and genetic alterations in such regions?

Answer 4.

The effect of non-imprinted genes located within the same genomic CNA as the imprinted loci could influence tumour development. In an attempt to understand the impact of additional genes we first looked for known tumour-suppressor and oncogenes mapping within the vicinity of the imprinted loci. In nine cases proven oncogenes mapped near to imprinted regions so that both loci could be co-amplified. Similarly, two imprinted regions mapped near tumour-suppressor genes that could be affected by the same deletions.

Imprinted region	Oncogene	Distance (bp)	Comment
PPIEL	MYCL1	373,144	
PLAGL1	BCLAF1	7,683,437	
GRB10	EGFR	4,286,685	Included in all amplification incorporating the centromere
MEST	MET	13,813,729	
PEG13	MYC	12,356,678	
INPP5Fv2	FGFR2	1,752,285	
H19	HRAS	1,480,856	Included in all amplification incorporating the telomere
MEG3	AKT1	3,937,929	Included in all amplification incorporating the telomere
MCTS2	BLC2L1	96,751	
Imprinted region	Tumour suppressor gene	Distance	Comment
DIRAS3	FOXD3	4,722,915	
NH2PL1	EP300	493,856	

As a general trend, the larger the distance between the cancer-associated gene and the imprinted loci the lower the frequency that both regions are involved in the same cytogenetic aberration. However in a high proportion of cases both loci are affected (i.e. *PPIEL* and *MYCL1* co-amplify in 98% of lung, 91% in liver, 97% in breast and 100% in colon cancer cell lines).

In addition to the classic acquired loss-of-heterozygosity associated with deletions, we show that *cnnLOH* is also a common chromosomal defect affecting imprinted loci. Since *cnnLOH* may lead to homozygosity of a pre-existing pathogenic mutations (i.e. silencing tumour-suppressor genes or activating oncogenes) we screened for genes with homozygous mutations mapping within *cnnLOH* regions harbouring imprinted loci. Of the 280 cell lines analysed, 258 had at least one region of *cnnLOH* affecting an imprinted locus (48 breast cancer line lines, 44 colon cancer cell lines, 14 liver cancer cell lines, 152 lung cancer cell lines) with

26% also containing homozygous mutated genes described from in the COSMIC database. The majority of mutated genes have not been associated with tumour initiation or progression, however we did identify reoccurring *RB1* mutations associated with cnnLOH of chr13q14.2 and *PTEN* mutation with cnnLOH of chr10q23-26 in lung and breast cancer cell lines.

Furthermore, the analysis of the parent-of-origin of the cnnLOH, as inferred by the methylation profile of the affected imprinted DMRs, suggests that the affected on imprinted gene dosage is complex. For example, cnnLOH for the chr11p15.5 interval, including *IGF2-H19*, involves the maternally and paternally-derived chromosomes equally in breast and lung cancer cell lines, but is exclusively paternal in colon cancer cell lines consistent with previous observations that *IGF2* over-expression is oncogenic. For cases also harboring mutated genes, it seems that the presence of the genetic variant is more influential than the parent-of-origin of the cnnLOH, with the exception of *RB1* mutations in lung cancer cell lines, in which all cases (7/7) were associated with hypermethylation at the *RB1* imprinted DMR.

A new results section has been added to the manuscript describing the associations between known cancer-associated gene and nearby imprinted loci. It reads “

The influence of nearby non-imprinted genes on CNAs

The effect of non-imprinted genes located within the same CNAs as the imprinted loci could also influence tumor development. In an attempt to understand the impact of additional genes we identified nine oncogenes and two tumor-suppressor genes that map within the vicinity of the imprinted loci (Supplementary Table 4). In general, the larger the distance between the cancer-associated gene and the imprinted loci the lower the frequency that both regions are involved in the same cytogenetic aberration. However, in many cases both loci are affected. For example, the imprinted *PPIEL* loci is located ~373 kb from the *MYCL1* oncogene and co-amplification of both regions was observed in 98% of lung, 91% in liver, 97% in breast and 100% in colon cancer cell lines.

In addition to the classic acquired LOH caused with deletions, we show that cnnLOH is also a common chromosomal defect affecting imprinted loci. Since cnnLOH may lead to homozygosity of pathogenic mutations (i.e. silencing tumor-suppressor genes or activating oncogenes) we screened for genes with homozygous mutations mapping within cnnLOH regions harbouring imprinted loci. Of the 280 cell lines analysed, 258 had at least one region of cnnLOH affecting an imprinted loci with 26% also containing homozygous mutated genes described from in the COSMIC database. The majority of mutated genes have not been associated with tumor initiation or progression. However we did identify reoccurring mutations of *RB1* associated with cnnLOH of chr13q14 and *PTEN* mutations with cnnLOH of chr10q23-26 in lung and breast cancer cell lines (Supplementary Table 5).

Further analysis of the parent-of-origin of the cnnLOH, as inferred by the methylation profile of the affected imprinted DMRs, suggests that the influence on imprinted gene dosage is complex. For example, cnnLOH for the chr11p15.5 interval involves both the maternally and paternally-derived chromosomes equally in breast and lung cancer-derived cell lines, but is exclusively paternal in colon cancer cell lines consistent with previous observations that *IGF2* over-expression is oncogenic (The Cancer Genome Atlas Network, 2012). For cases also harboring homozygous

mutated genes, it seems that the presence of the genetic variant is more influential than the parent-of-origin of the cnnLOH, with the exception that *RB1*, in which all lung cancer cell lines (7/7) were associated with hypermethylation at the *RB1* imprinted DMR”.

Furthermore an two additional tables have been added to the supplementary information describing the frequency of co-amplification or co-deletion incorporating cancer genes and imprinted loci (Supplementary Table 4) and a list of cnnLOH regions incorporating imprinted loci associated homozygously mutated genes (Supplementary Table 5).

REVIEWER 2.

Answer to general comments.

We thank this reviewer for their suggestions. Unfortunately due to manuscript reformatting (i.e shortening the title to only 15 words and cutting the abstract from 250 to 150 words) we are unable to specify that we analysed four human cancer types.

Question 1.

It would be helpful if the authors could clarify at which imprinted loci there were no apparent relationship between CNAs and the parent-of-origin of the amplified/deleted chromosome and which loci (and tumor types) there appeared to be a relationship.

Answer 1.

We have now included an additional supplementary figure representing a summary metric for each imprinted region / cancer type that signifies the estimated the proportion of methylation variability explained by copy-number. A sentence has been added to the manuscript that reads, “An estimate of the proportion of methylation variability explained by copy-number alone is shown in the supplementary information (Supplementary Fig. 2).” In addition we have included a brief description of the analysis in the methods section.

REVIEWER 3.

We thank the reviewer for their constructive comments.

Question 1.

Imprinted loci that are amplified to high levels maybe those that are close to focally amplified oncogenes. For example PPIEL is ~300 kb from the oncogene MYCL1. The authors should comment on the frequency of these occurrences.

Answer 1.

Please see our answer to reviewer 1 question 4.

Question 2.

Figure 1A – It would be helpful if the loci names were appended with chromosomal

arm locations so that you could compare this data to the Sky karyotype in figure 1B. The labels of copy number to the left of MDA-MB453 are too small to be seen. Is the SNURF locus in Figure 1A the same as the SNRPN locus in Figure 1D? If so the same name should be used throughout the manuscript.

Answer 2.

We have ammended Fig.1 as suggested. Unfortunately due to space restrictions we could not label the imprinted loci on the SKYE karyotype. However we have now included fully labelled chromosome ideograms on our lab webpage that accompanies the supplementary information.

The *SNRPN* gene has many transcript isoforms with the DMR regualting the imprinting throughout the 15q region located within the promoter of the specific isoform names *SNURF* (SNRPN Upstream Reading Frame). Therefore the nomenclature for the locus and specific DMR are correct.

Question 3.

The authors claim that for chromosomes harboring more than one imprinted domain, each locus is associated with private focal CNAs. However in figure 1D, it appears that in ~25% of cell lines the SNRPN and IGF1R loci are co-amplified due to a chromosomal arm level gain of 15q. These amplifications are neither private nor focal CNAs.

Answer 3.

We thank the reviewer for spotting this mistake. We have now ammended the sentence to reads "For chromosomes harboring more than one imprinted domain, the CNAs may be focal or alterations involving the entire chromosome arm."

Question 4.

Copy number is not interpretable without understanding the baseline ploidy of tumours. For instance a segment with a copy number of 3 is an amplification in a diploid tumour but a deletion in a tetraploid tumour that has undergone whole genome doubling (MDA-MB-453 is an example of this). The authors use of copy ratios shows a good understanding of this issue. But if tumours shown in figure 1D are of different baseline ploidy than copy number should be shown relative to the baseline ploidy (total alleles/baseline ploidy). Copy number shown in tables also should include baseline ploidy or relative copy number.

Answer 4.

We fully agree with the reviewer that understanding baseline ploidy of the tumours is important. We have therefore generated new tables (Supplementary Table 1) and figures (Supplementary Fig. 1) showing the relative baseline ploidy (total alleles/relative copy-number). In addition we have included a sentence in the results section addressing CNAs at imprinted domains that reads, "In all cases an estimated ploidy baseline (total alleles/baseline ploidy) was also calculated (Supplementary Table 1; Supplementary Fig. 1) since total copy number >2 could represent amplification in a diploid tumor but a deletion in a hyperploid tumor "

Question 5.

Copy number changes shown in tables should also give some indication on whether these changes are due to focal or arm level alterations. One way of doing this is to indicate what % of the total chromosomal arm do these alterations represent.

Answer 5.

To describe the precise size of the deletions and amplifications for each affected imprinted region in all cancer cell lines would result in an enormous supplementary table that would be extremely difficult to use. However to address this comment we have generated maps of all cytogenetic aberrations for each imprinted region for the cell lines from four cancer types (the same as Fig. 1D) which are available on our laboratories webpage.

Question 6.

Figure 2C is difficult to understand and should have a legend. It is unclear what the black and white circles underneath methylation maps of loci are supposed to represent.

Answer 6.

For clarity we have amended the legend for Fig. 2C, it now reads "Two different bisulphite PCRs were performed per region to confirm the strand-specific methylation profile as determined by cloning and direct sequencing. Each circle represents a single CpG dinucleotide on a DNA strand (results for multiple DNA strands are depicted as rows), filled circles indicate a methylated cytosine, and open circles an unmethylated cytosine."

Question 7.

In figure 3B, it is unclear what exactly the X axes of the graphs are.

Answer 7.

The x-axis on the graphs in Figures 3B and 5C represents the number of cell lines with with cancer-associated methylation changes. We have amended the figure and legend that now reads "The left column graphs reveal that cell lines with the highest hypermethylation burden for CIMP regions are similarly hypermethylated at bivalent domains. The middle column is a comparison between the methylation profiles of imprinted DMRs irrespective of CNA status and CIMP. The right row is the same comparison but with only imprinted domains with a normal copy-number. For each type of loci the number of genes analysed is indicated on the X-axis."

Question 8.

In figure 4B the RNA FISH signals for cell line MCF7 are barely visible. Can you show a better representative cell for this figure?

Answer 8.

The intensity of the RNA FISH signals for the ncRNA KCNQ1OT1 is quantitative therefore we are reluctant to manipulate the images. To make the RNA FISH signals more visible we have now included a zoomed in insert panels in Fig. 4B and

supplementary Fig. 5. The figure legends now read “Representative RNA-FISH analysis of *KCNQ1OT1* lncRNA-coated territory (green signal, white arrows) of individual nuclei with inserts representing zoomed in images of FISH signals”.

Question 9.

For figure 5B the same comment as figure 1D. If the data in 5B is not relative copy number than how did the authors take in account differences in tumour purity when calculating copy number?

Answer 9.

All CNAs depicted in the figure are relative copy-number and tumour-purity was not taken into account. The SNP array data used for copy-number calling was processed data available for TCGA, therefore any corrections would be impossible to perform. The same is true for the methylation analysis and it for this reason that we included supplementary table 4 describing the tumour characteristics.

REVIEWERS' COMMENTS:

Reviewer #1 (Remarks to the Author):

The authors have provided a thorough rebuttal. I have no further comments.

Reviewer #3 (Remarks to the Author):

I am satisfied with the changes that the authors made, and I think that the manuscript is now suitable for publication.